# Coupled and Coordinated Analysis of Urban Green Development and Ecological Civilization Construction in the Yangtze River Delta Region

**Xinyu Hu [1], Chun Dong [2,\*] and Yihan Wang [1]**

[1] School of Geomatics, Liaoning Technical University, Fuxin 123000, China
[2] Chinese Academy of Surveying and Mapping, Beijing 100039, China
\* Correspondence: dongchun@casm.ac.cn; Tel.: +86-139-1198-3769

**Abstract:** Managing the human–nature relationship is key to facilitating the sustainable development of cities. The coupled coordination relationship between ecological civilization construction and urban green development and influence of spatio-temporal heterogeneity has been insufficiently studied. We used the coupled coordination degree model (CCDM) and spatio-temporal weighted model (GTWR) to analyze the relationship and heterogeneity between ecological civilization construction and UGD and ECC in each city in the Yangtze River Delta region from 2010 to 2019. The results show that: (1) UGD and ECC coordination levels fluctuated more from 2010 until 2019. There was a transition from lagging UGD and ECC to lagging ecological civilization construction and a decreasing degree of coupling coordination in the Yangtze River Delta region from east to west from near imbalance to primary coordination. (2) The Yangtze River Delta's negative UGD and ECC effect was concentrated in northwest inland cities; the positive UGD and ECC effect was concentrated in southeast coastal cities. Thus, UGD and ECC and ecological civilization construction complement each other. This study provides a scientific basis for analyzing the coordination between ecological civilization construction and UGD and ECC and provides practical guidance for formulating and implementing urban high-quality development countermeasures.

**Keywords:** Yangtze River Delta region; ecological civilization construction; urban green development; coupled coordination degree model; spatio-temporal weighted regression model



## 1. Introduction

The 20th Party Congress report of the Chinese Communist Party outlined specific requirements for the creation of novel concepts that "enhance system for building ecological civilization, accelerate urban green growth, and promote to live in harmony of humans and nature" [1]. The Party and State attach a high level of importance to the promotion of an ecological civilization by the municipal government. This has highlighted the new strategic concepts behind the new development pattern of the double cycle of cities, and the positive contribution to socio-economic development and sustainable development. Li [2] has defined ecological civilization as a natural ecological and environmental civilization that depends on the exchange of human knowledge and wisdom, relies on a healthy ecological and natural balance, and coordinates urban, social, and economic growth with overall growth of the global economy. According to Gu [3], an ecological civilization is a high-quality culture that reflects the harmonic relationship between humans and nature, constitutes modern human civilization, and has strong contemporary relevance. Cities are the primary site for ecological civilization construction (ECC), and urban green development (UGD) should prioritize ECC, promote urban informatization, greening, and sustainable development. This should be undertaken through green urban economic growth and attach importance to the coordinated development of the urban economy, society, and culture. It should also emphasize resource-intensive industries to raise environmental awareness. In terms of

layout and direction, UGD is a long-term driving force for building ecological civilization. Therefore, studying the interaction mechanism and evolution law of UGD and ECC has become a popular research topic. Many studies have investigated the relationship between UGD and ECC from various perspectives, including urbanization [4], e-commerce [5], economic development [6], energy consumption [7], and environmental pollution [8,9]. These empirical studies provide important references for further exploring the coupled and coordinated relationship between UGD and ECC. However, most prior studies in the field have primarily focused on the qualitative relationship between UGD and ECC and the study of the coordination within a single system, while disregarding the coordination relationship and inner mechanism between UGD and ECC. To date, there has been relatively little research on the coupling connection between UGD and ECC from a quantitative perspective of the composite system. This is not conducive to in-depth exploration of the interaction and degree of influence between urban–ecological civilization.

The term "ecological civilization" was first proposed by Professor Iring Fetscher [10] in Germany. However, no relevant definition was given in his work. For the exploration of the connotation of ecological civilization, foreign scholars have conducted relevant studies from multiple levels, perspectives and multidisciplinary integration, such as exploring the relationship between Marxist ideology and ecological civilization [11], building global democracy to develop ecological civilization [12], establishing an ecological economy to explore the roots of ecological civilization [13], improving the political economy framework to pursue practical steps towards ecological civilization [14], and building a global ecological civilization system by creating new paths of sustainable development to replace traditional economic development models [15]. The idea of green development began to emerge from abroad when Rachel Carson [16] argued in the *Silence of Spring* that the natural world on which we depend is being destroyed by man's uncontrolled use of chemical pesticides. This was the first time that man stood up for man and nature and revealed the negative effects of scientific progress. In 1970, an eco-justice campaign on the theme of 'ecological economic value theory and sustainable development theory' erupted in the USA, revealing a shift towards the practice of green development, as reflected in ecological economic value theory, and emphasizing how the ecological core of green development is a solution to environmental protection through sustainable development rather than simple confrontation [17]. The concept of green development has caused considerable controversy in the West. This concept has caused quite a stir in Western countries. Starting in the 1990s, the global green movement began to emerge, from 'green production' to 'green consumption' [18,19], from "from 'green logistics' to 'green distribution' [20,21], from 'green technologies' to 'green institutions' [22,23], from "green development" to "green civilization" [24,25], the concept of green development has gradually been integrated into the political sphere and has played an important role in promoting the 'greening' of the platforms of ruling parties around the world. Although research on green development and ecological civilization has developed rapidly in recent years abroad, a systematic theoretical system has not been formed for green development, and theories, such as green economy and ecological civilization, are uncertain in foreign countries, with no mature experiences and models, and insufficient academic research.

In China, as the pioneer of research on "ecological civilization", Ye Qianji developed the notion of "China's ecological civilization" as early as 1987 at a national seminar on "How to realize ecological development of agriculture" and related issues. He conducted an in-depth study of related concepts and published in his book "Ecological Agriculture: The Future of Agriculture" [26]. The book discusses the direction of agricultural development in the context of China's national conditions and focuses on the theory of ecological agriculture with Chinese characteristics. After the Party and government continuously improved the relevant concepts, the 18th Party Congress formally put forward the concept of ECC, with the principle aims of promoting sustainable development, to plant trees for future generations to "take advantage of the cool", and not deprive future generations of resources and ecological heritage [27]. The ECC is an important component of Chinese socialism, closely linked to the

welfare of the populace, the nation's future, the "two hundred years" goal, and the Chinese aspiration to realize the great rebirth of the Chinese nation. Meanwhile, UGD serves as the core and pillartion and the fundamental principles of socialist core values, highlighting that UGD is a vivid practice and detailed carrier of the ECC [28]. As a result, the two have an interactive coupling relationship. In recent years, many scholars have begun to pay attention to the relationship between UGD and ECC. The relevant research mainly includes the following three aspects: first, exploring the connotation theory of UGD and ECC [29]; the second is to construct an indicator system for UGD and ECC, and comprehensively evaluate it using entropy weight TOPSIS method [30,31], analytic hierarchy process [32,33], and comprehensive evaluation method [34]; the third is to reveal the coupling and coordination relationship between UGD and ECC [6,35,36]. However, there are two areas where the current research needs to be strengthened. When it comes to indicator choice, there are relatively few studies that have quantitatively examined the relationship between UGD and ECC from an integrated system of economic, social, cultural, and environmental perspectives. Meanwhile, UGD is a multidimensional process involving economics, production, and life [37–39]. Prior research has typically concentrated on a single aspect of UGD while neglecting other socioeconomic and other indicators, making it more challenging to determine the level of UGD. Most studies have concentrated on the qualitative effects of UGD on the ECC, primarily studying the relationship from a social science perspective, excluding the spatial and temporal characteristics, and collaborative coupling between UGD and ECC. This makes it difficult to study the inner workings of UGD and ECC. The coupled coordination degree model (CCDM) has been previously used extensively to describe the overall effectiveness and coordination effects between composite systems. However, it cannot be used to look at how these two horizontal systems affect each other and how they change over time from the perspective of spatio-temporal heterogeneity. Therefore, a spatio-temporal model needs to be introduced to analyze specific influencing factors.

According to Tobler's first law of geography, any geographic feature or property is connected to others in space [40]. Modifications to the local spatial attributes will considerably affect the neighborhood. To avoid bias in the model assessment results, it is crucial to account for spatial dependence when examining the impact mechanisms [41]. Prior exploration of the influencing factors of ECC has not considered that the components of ECC are interrelated and influenced by economic development, environmental protection, and urban construction, and have shown pronounced spatial spillover effects [30,42,43]. Meanwhile, in the UGD process, production inputs, such as labor, industry, and capital, are transported between areas, which indicates that the creation of local ecological civilization may be influenced by the green growth of neighboring cities. Currently, scholars have recognized this issue and have considered spatial non-stationary correlations based on ordinary least squares (OLS) regression. GWR models are used to effectively reflect the spatial relationships between various variables by establishing local regression equations, which can provide positional guidance for decision making [44]. However, variations in how economic growth and urban infrastructure development affected the degree of ECC at various stages of UGD have also been identified [45,46]. This has indicated that the relationship between UGD and ECC fluctuates over time; indeed, time is a key factor that cannot be disregarded. To this end, this paper utilizes a spatio-temporal geographically weighted model (GTWR) that takes into account the temporal factor in order to better explain the potential relationships between UGD and ECC variables from the perspective of spatio-temporal heterogeneity, which greatly improves the accuracy of model simulations.

This study has examined the spatio-temporal variability among UGD and ECC of cities in the Yangtze River Delta area (YRD) from 2010 to 2019 by coupling CCDM and GTWR models. The following two factors led to the YRD being chosen as the study region: ① Given the YRD's increasing population density and resource exploitation intensity, the government has accelerated urbanization through high levels of industrial development to meet the requirements of economic growth. This has exceeded the YRD's ecological and environmental carrying capacity, resulting in numerous ecologic and environmental

problems. Therefore, policymakers face a new task in determining how to balance the relationship between urban expansion and environment conservation. ② The YRD is an important birthplace of Xi Jinping's idea of ecological civilization, and the core value of "green water and green mountains are golden mountains" is deeply ingrained in people's hearts. The YRD has a dense network of water and rich ecosystem types, with mountains, water, forests, fields, lakes, and seas complementing each another. The natural ecological environment is one of the main components of the national policy for integrated and sustainable development of the YRD. Green and environmentally friendly development is the key solution for the YRD integration's outstanding natural ecological environment concerns, and to promote high-quality and long-term development of the regional economy.

There are two main innovations in this study. ① Methodological innovation: A new GTWR approach has been developed to investigate the interaction of UGD and ECC. This not only successfully extends the range of applications for GTWR, but also considers the relevance, diversity, and spatio-temporal heterogeneity of UGD and ECC. ② Perspective innovation: unlike prior qualitative research that has only concentrated on the political and value importance of UGD's impact on the ECC, this study used CCDM to investigate the coupled and coordinated interaction between UGD and ECC from a quantitative perspective of composite systems. The main contributions of this paper are as follows: ① The main influencing factors of UGD and ECC in the YRD region are analyzed, on the basis of which a comprehensive evaluation index system of UGD and ECC is constructed. ② Combining CCDM and the entropy-weighted TOPSIS method, a coupled and coordinated relationship model of UGD and ECC is proposed. The entropy-weighted TOPSIS method is able to evaluate the indicator data objectively and effectively avoid the errors brought by the evaluation of subjective factors. ③ Based on the GTWR model, the spatio-temporal heterogeneity of UGD and ECC in the YRD region is analyzed, which helps to identify the intrinsic correlation mechanism between UGD and ECC, put forward policy recommendations to promote green development and ecological civilization in cities in the YRD region, and provide a basis for policy makers to realize the high-quality development of ecological cities.

## 2. Study Area, Data Sources, and Conceptual Framework

### 2.1. Study Area

The YRD is a natural geographical area comprising of a delta progressively generated by the Yangtze and Qiantang rivers interacting with the sea [47]. The area includes 41 prefecture-level cities in Anhui, Jiangsu, Zhejiang, and Shanghai, the location map of the study area is shown in Figure 1. The YRD encompasses approximately 358,000 square kilometers, accounting for 3.7% of China's national territory. By the end of 2019, the population size accounted for 6% of the entire population, and the economic size represented 25% of the country's total economic volume. This has made it one of China's regions with the most integral capabilities, the highest level of population density, and the most developed economy. However, given the high population density of the YRD, intensive resource use, and level of urbanization, the natural biological environment there faces a number of regional and structural issues. The YRD and Huai River watershed systems have not been sufficiently protected, and relatively intact ecological space has been occupied in some areas. Environmental risks and hidden dangers are prominent from the petrochemical industry present along the rivers. The total resource and energy costs and pollutant emissions in the region are high, air quality indicators have not been met, and the pressure from peak greenhouse gas emissions is excessive.

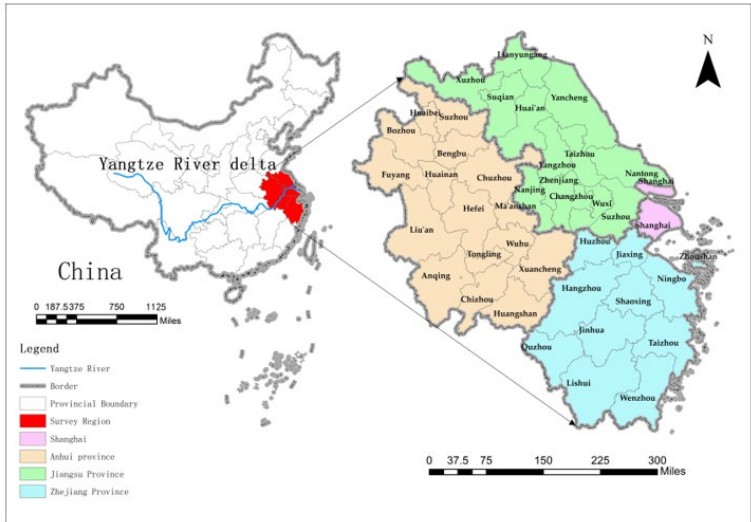

**Figure 1.** Location map of the study area. Note: this map is based on the standard map with the approval number GS(2020)4619 downloaded from the standard map service website of the Ministry of Natural Resources, and the base map has not been modified.

### 2.2. Data Sources

The data presented in this study have drawn on panel data from 41 local administrative regions in the YRD from 2010 to 2019. Panel data were derived from national, provincial, and municipal government statistical yearbooks, such as the *China City Statistical Yearbook*, *China Regional Economic Statistical Yearbook*, *Anhui Statistical Yearbook*, and the *Jiangsu Statistical Yearbook*, *Zhejiang Statistical Yearbook*, as well as statistical bulletins on each city's social development. Some missing data could be supplemented with the help of least squares or numerical smoothing. Considering the different positive and negative directions and unit magnitudes of each basic index data, before analyzing the data, it was necessary to standardize the raw data using the formula for positive index standardization (1) and the formula for negative index standardization (2) with the following formulas:

$$b_{ij} = \frac{a_{ij} - a_j^{min}}{a_j^{max} - a_j^{min}} \qquad (1)$$

$$b_{ij} = \frac{a_j^{max} - a_{ij}}{a_j^{max} - a_j^{min}} \qquad (2)$$

where $b_{ij}$ is the standardized value and $a_{ij}$ is the ith value of the jth indicator; $a_j^{max}$ and $a_j^{min}$ are the maximum and minimum values of the j indicators.

### 2.3. Conceptual Framework

There is a complex interaction between UGD and ECC. Ecological civilization and UGD are two dimensions of sustainable development, both of which have been proposed in response to the real dilemmas faced by people, such as resource depletion, energy shortage, global warming, ecological environment deterioration, and urban ecological pollution. The ECC solves the problem at the ideological and strategic levels, and UGD solves the problem at the level of strategic measures. This is ultimately implemented and developed through the combination of people, industry, and economy. Promoting a life in tune with nature is the most important aspect of promoting China's ECC. It is crucial to deal with the peaceful coexistence of people and nature, solve outstanding environmental problems with a value-based approach, promote the steadfast fight against pollution, increase the effectiveness of resource allocation, and raise the bar on urban development quality to guarantee sustainable development of the national economy and society. A

strategic step toward achieving ECC, green and sustainable urban development is also an unavoidable decision for ecological civilization. The first aim of UGD is strategic and scientific green urban planning. This is then followed by improvement and optimization of green industrial structures and, ultimately, green technical innovation. Fundamentally, effective management of the relationship between humans and nature is the issue that ecological civilization and UGD are meant to tackle. Human lifestyles will influence the natural ecological environment because humans are a part of nature. As a result, we need to develop basic ideals on how to live in harmony with nature, observe natural laws, and safeguard natural resources. Industrial growth is regarded as an essential engine of urban expansion, with considerable implications for the natural biological environment. In this process, it is necessary to establish a sense of "ecological rationality" and consciously consider the ECC as a strategic guiding ideology. UGD should combine the elements of space, new energy, new industry, new technology, new themes, and other elements organically. This then promotes the black industrial development mode of "overspending resources and destroying the natural ecological environment" to transform into the green sustainable development choice mode. The transformation, coupling and coordination of multiple elements of "people–industry–economy" enable cities to achieve green growth in the economic system, increase green wealth in the industrial production system, and further enhance green welfare in the social life system. Determining the coordination relationship and internal workings of UGD and ECC can therefore support the development of effective policies for high-quality urban development in a conceptual framework, as shown in Figure 2 below.

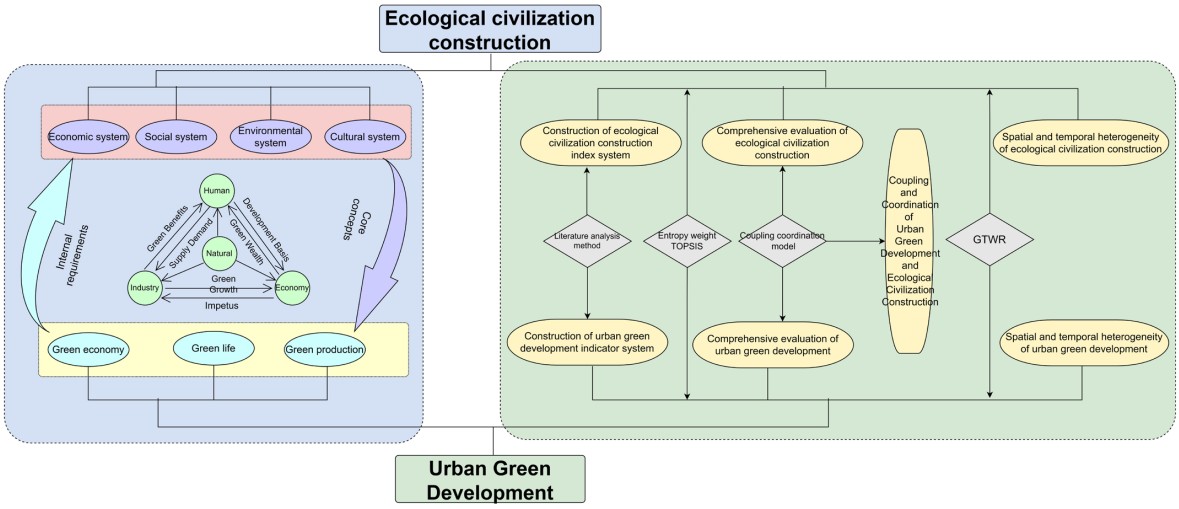

**Figure 2.** Main research methods and processes.

## 3. Research Methodology

The following steps make up the research methodology and main procedure employed in this article, which are depicted in Figure 2. ① Appropriate assessment indicators for UGD and ECC are studied through a literature analysis method. ② The comprehensive evaluation index of UGD and ECC was calculated using the entropy-weighted TOPSIS approach. ③ The degree of coupled coordination among UGD and ECC within YRD prefecture-level cities was examined using the CCDM. ④ Based on the GTWR model to explore the spatio-temporal heterogeneity among UGD and ECC. Here are some specific research methods:

### 3.1. UGD and ECC Evaluation: Based on an Entropy-Weighted TOPSIS Method Measure
3.1.1. Indicator System Construction

The foundation for understanding the coupled and coordinated relations between the ECC and UGD is the scientific and thorough creation of the related evaluation index system.

The evaluation index of ECC predominantly refers to the National ECC Assessment Target System (revised in 2016) and draws on existing research results [13,15,48–51]. It combines the characteristics of the YRD, according to the systematic, representativeness and usability principles. The ECC is divided into the four primary indicators of the economic system, cultural system, social system, and the environmental system. A total of 20 secondary indicators, such as the rate of forest cover and urban population density, are used to construct an index system for measuring the ECC level in the YRD. The evaluation index of ECC is used to characterize the ECC level of each prefecture-level city in the YRD. The UGD evaluation index system mainly refers to the UGD evaluation indices in the National Green Development Index System (revised in 2016). It also draws on other relevant research results [14,37,38,52,53] from the green economy, green production, and green life. The 16 indicators, such as the proportion of employees in the secondary industry and industrial wastewater emissions, were selected from three first-level indicators to construct a detailed evaluation index for UGD to characterize the UGD level of each city at the prefecture level in the YRD. Because there may be multicollinearity among the variables selected according to a literature review, this paper uses variance expansion factor (VIF) to analyze multicollinearity of all variables before spatial econometric analysis. The results show that the VIF of all variables is less than 10, so there is no multicollinearity between variables, and regression analysis can be carried out. The operability and accessibility of index data were taken into consideration when building the coupled and coordinated assessment index system of UGD and ECC in the YRD, as shown in Table 1.

**Table 1.** Coupled and coordinated evaluation index system of ECC and UGD in the YRD.

| Primary Index | Secondary Index | Three-Level Index | Unit | Characteristic | Weights | VIF |
|---|---|---|---|---|---|---|
| ECC | Economic System | GDP percentage for the tertiary sector | % | + | 0.029 | 3.41 |
| | | Energy usage as a percentage of GDP | Ton of common coal for every million yuan | − | 0.013 | 1.749 |
| | | Fiscal revenue as a share of GDP | % | + | 0.041 | 1.55 |
| | | Income available per person in urban households | Yuan | + | 0.048 | 4.38 |
| | | Rural dwellers' disposable income per capita | Yuan | + | 0.051 | 2.103 |
| | Cultural System | Number of students registered at institutions of higher learning in general | People | + | 0.124 | 2.047 |
| | | Public library book collection | Thousands of copies, pieces | + | 0.198 | 6.473 |
| | | Number of full-time teachers in higher education | People | + | 0.140 | 3.562 |
| | | Education spending as a share of GDP | % | + | 0.044 | 1.172 |
| | Social Systems | Urbanization rate | % | + | 0.028 | 2.941 |
| | | Urban population density | (People/km$^2$) | − | 0.006 | 2.128 |
| | | Number of doctors (practicing physicians and practicing assistant physicians) | People | + | 0.066 | 2.496 |
| | | Urban medical insurance coverage rate | % | + | 0.044 | 2.461 |
| | | Urban pension insurance coverage | % | + | 0.043 | 1.215 |
| | | Urban Engel coefficient | | − | 0.018 | 1.902 |
| | Environmental Systems | Coverage of forests | % | + | 0.063 | 1.199 |
| | | Greening coverage of built-up areas | % | + | 0.016 | 1.065 |
| | | Industrial wastewater discharge | Million tons | − | 0.009 | 3.895 |
| | | Comprehensive use rate for general industrial solid waste | % | + | 0.009 | 1.077 |
| | | Industrial sulfur dioxide emissions | Ton | − | 0.008 | 4.209 |

**Table 1.** *Cont.*

| Primary Index | Secondary Index | Three-Level Index | Unit | Characteristic | Weights | VIF |
|---|---|---|---|---|---|---|
| Urban Green Development | Green Economy | The percentage of personnel in secondary industry | % | − | 0.05 | 7.574 |
| | | The percentage of workers in tertiary industry | % | + | 0.05 | 6.75 |
| | | Total labor productivity | 10,000 Yuan/person | + | 0.04 | 1.111 |
| | | Food crop production | Million tons | + | 0.11 | 1.061 |
| | | Retail sales of social consumer products | Billion | + | 0.18 | 1.036 |
| | Green Living | Resident population | 10,000 people | − | 0.01 | 2.471 |
| | | Green space per capita | m$^2$ | + | 0.10 | 6.754 |
| | | Road area per capita | m$^2$ | + | 0.10 | 5.425 |
| | | Sewage treatment rate | % | + | 0.03 | 1.217 |
| | | Garbage disposal rate | % | + | 0.02 | 1.182 |
| | | Share of urban construction land | % | − | 0.01 | 3.398 |
| | Green Production | Discharge of industrial wastewater | Million tons | − | 0.01 | 7.01 |
| | | Industrial emissions of sulfur dioxide | Ton | − | 0.02 | 4.333 |
| | | Emissions of industrial fumes | Ton | − | 0.03 | 1.915 |
| | | Investment in environmental management as a percentage of GDP | % | + | 0.09 | 1.181 |
| | | Patent for inventions in the environmental protection industry | Piece, individual | + | 0.16 | 4.926 |

3.1.2. Entropy Power TOPSIS Method

The entropy-weighted TOPSIS approach combines the entropy-weighted method and the TOPSIS method to provide a full examination of the fundamental indicators. When measuring the system of comprehensive evaluation indices, the subjective assignment method and the objective assignment method are the most common methods used to determine the weight index system. The entropy TOPSIS method belongs to the objective assignment method. The indicators were objectively weighted based on the degree of variation in each indicator value to avoid the influence of subjective factors, calculating the distance between each appraisal method and the best and worse scheme. The relative overlap between each evaluation scheme and the best scheme was then determined, and finally the evaluation schemes were ranked. The indicator weights and the final evaluation indices were calculated from the indicators. The evaluation outcomes were improved by using the entropy weight TOPSIS approach, and the specific calculation methods are as follows.

① Building a standardized matrix:

$$A = \left(a_{ij}\right)_{m \times n} \tag{3}$$

where $a_{ij}$ represents the data after normalization.

② Determine the entropy of evaluation indicators:

$$f_{ij} = a_{ij} / \sum_{i=1}^{m} a_{ij}$$
$$H_j = -\frac{1}{\ln m} \left( \sum_{i=1}^{m} f_{ij} \times \ln f_{ij} \right) \tag{4}$$

where $f_{ij}$ denotes that the first $j$ indicator in the first $i$; the weight of the indicator in the first program, and the weight of the indicator in $H_j$ denotes the entropy value of the $j$ the entropy value of the indicator.

③ Calculate the entropy weight of the index:

$$w_j = \left(1 - H_j\right)/\sum_{j=1}^{n}\left(1 - H_j\right) \tag{5}$$

where $w_j$ denotes the first $j$ weight of the indicator.

④ Calculation of weighted evaluation weights:

$$R = \left(r_{ij}\right)_{m \times n}, \text{ where } r_{ij} = w_j a_{ij} \tag{6}$$

where $r_{ij}$ denotes the weighted standardized index value.

⑤ Determine the optimal solution $Q_+$ and the inferior solution $Q_-$:

$$\begin{aligned} Q_+ &= \left(r_1^+, r_2^+, \cdots, r_n^+\right) \\ Q_- &= \left(r_1^-, r_2^-, \cdots, r_n^-\right) \end{aligned} \tag{7}$$

⑥ Calculate the distances of each program from $Q_+$ and $Q_-$:

$$\begin{aligned} D_i^+ &= \sqrt{\sum_{j=1}^{n}\left(r_{ij} - r_i^+\right)^2} \\ D_i^- &= \sqrt{\sum_{j=1}^{n}\left(r_{ij} - r_i^+\right)^2} \end{aligned} \tag{8}$$

⑦ Calculate the evaluation index of each program:

$$C_i = \frac{D_i^-}{D_i^- + D_i^+} \tag{9}$$

where $C_i \in [0,1]$, and the higher the score of the city $C_i$, the higher the level of green development and ECC of the city.

### 3.2. Coupled Coordination Degree Model (CCDM)

The coupling degree is a concept that originates from physics but has wide application in other research fields given the relatively similar coupling links between systems. It works by analyzing how closely two or more systems interact and are coupled, which establishes an ordered trend in integrated systems. The degree of coupling (Formula (10)) and the degree of coupling coordination (Formulas (11) and (12)) were used in this study to measure and assess the degree of coordination between the UGD and ECC and its subsystems. The following are the specific models:

$$C = \sqrt{(U_1 \times U_2)/((U_1 + U_2)/2)^2} \tag{10}$$

$$T = \alpha U_1 + \beta U_2 \tag{11}$$

$$CD = \sqrt{C \times T} \tag{12}$$

where $C$ stands for the level of coupling between the UGD and ECC, $T$ represents the overall evaluation index, and $CD$ represents the level of coupling coordination between the UGD and ECC. The comprehensive indices of UGD and ECC are denoted by the letters $U_1$ and $U_2$, respectively. Given that these two subsystems are equally crucial for determining the level of coordination amongst UGD and ECC, they are both assigned the same weight, or 0.5 [54]. In this study, referring to the classification of coordination types in the related research literature [4,55], the coupling coordination types of UGD and ECC are classified into four categories on the basis of the magnitude of the coupling coordination degree $CD$, that is, when $0 < CD \leq 0.3$, it indicates no correlation. When $0.3 < CD \leq 0.5$, it indicates

a low degree of correlation, when $0.5 < CD \leq 0.8$, it indicates primary coordination, and when $0.8 < CD \leq 1$ indicates quality coordination.

### 3.3. Space–Time Weighted Regression Model (GTWR)

Wang et al. [56] highlighted that "although geographically weighted regression (GWR) models can deal with spatial heterogeneity and detect spatially varying patterns of relevant variables, they do not directly consider spatial dependence, and their residuals are uncertain". The spatio-temporal GTWR proposed by Huang Bo et al. [57] incorporates the spatio-temporal characteristics of the data into the regression model based on the GWR model, that is, it incorporates spatio-temporal distances and constructs (*X,Y,T*), and three-dimensional coordinates, which can simultaneously consider the effects of spatio-temporal distances on each explanatory variable. As a result, the spatio-temporal variability among UGD and ECC was examined in this article using the GTWR model in the following manner:

$$Y_i = \beta_0(\mu_i,\ v_i,\ t_i) + \sum_{k=1}^{p} \beta_k(\mu_i, v_i, t_i) X_{ik} + \varepsilon_i; i = 1, 2, \ldots, n \tag{13}$$

where $Y_i$ is the value of the explanatory variable for the *i*th city, $n$ is the number of cities in the YRD, and $k$ is the number of explanatory variables for the *i*th city; $t_i$ is the temporal coordinate of the *i*th city. $\beta_0(\mu_i, v_i, t_i)$ denotes the spatio-temporal intercept term for city I; $X_{ik}$ denotes the value of the *k*th explanatory variable for city I; $\beta_k(\mu_i, v_i, t_i)$ denotes the regression coefficient of the *k*th explanatory variable for city I as a function of the spatio-temporal coordinates; and $\varepsilon_i$ is the error term.

## 4. Analysis of Results

### 4.1. Analysis of the Spatial and Temporal Evolution Pattern of UGD and ECC

4.1.1. Time Evolution Pattern Analysis

Since the YRD's regional integration strategy was upgraded to a national stratagem in 2018, emphasizing ecological protection and promoting a green economy as a priority for cities in the YRD, and enhancing economic development, industrial production, and people's lives, the level of UGD in the YRD has shown a gradual upward trend from 2010 to 2019 (Figure 3). Since 2018, UGD has shown a rapid uptrend trend. Green economic development from 2010–2019 showed an upward trend in the early period (2010–2012), and the middle and late period (2012–2019) showed a changing trend from a stable period to a fluctuating upward trend. However, it was always maintained at a low level. Green production development has shown a fluctuating upward trend from 2010–2018, fluctuating even more in this time period in the technology mapping stage. After 2018, city green development with an upward trend has shown that green production can promote UGD. Green living as a whole shows a declining trend. The reason is that from 2005–2015, due to the YRD region's vigorous development of urban economy resulting in extremely rapid population growth, the population density grew too fast and the urban infrastructure construction could not meet the negative impact brought by the rapid population, so green life as a whole showed a decreasing trend from 2010–2019, but green life was always at a higher level, indicating that green life provides a strong impetus to UGD. This demonstrates that in the early stages of green city development in the YRD (2010–2017), when UGD was relatively slow, it was in the exploration stage. In the later stages of green city development (2018–2019), green production development and green living development gradually took over as the primary driving forces for the UGD.

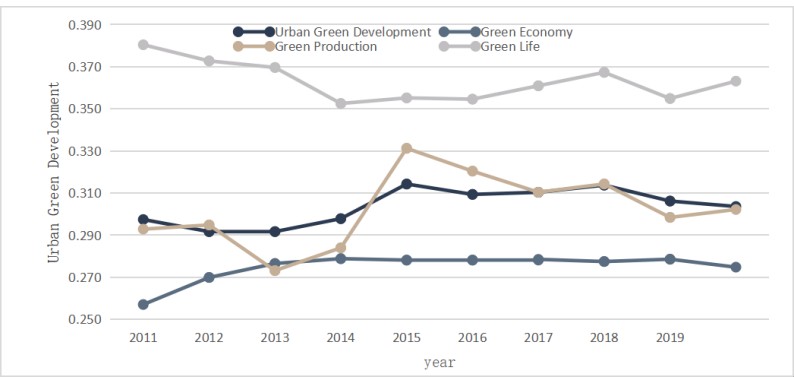

**Figure 3.** Temporal evolution pattern of UGD.

Figure 4 shows the trends of ECC and its four subsystems in the YRD from 2010 to 2019, in which the level of ECC shows a slow upward tendency. The level of ECC has progressively improved during this time, as shown by the increase in mean ECC value from 2010 to 2019 and from 0.235 to 0.268 in 2019. The trend of environmental system change has shown a slow decline followed by a fluctuating increase. This has indicated that since 2013, The YRD has improved environmental protection efficacy and supported the advancement of ecological civilization level. The similarities between the change patterns of the economic and cultural systems and those of the ECC have shown that these two systems are crucial to the process of enhancing the ECC. Building the social system is crucial in advancing ECC given that it was in a rapidly increasing trend prior to 2014. However, it remained relatively constant after 2014 and always remained at a relatively high level. This has shown that the lives of people in the YRD have generally been maintained at a high level.

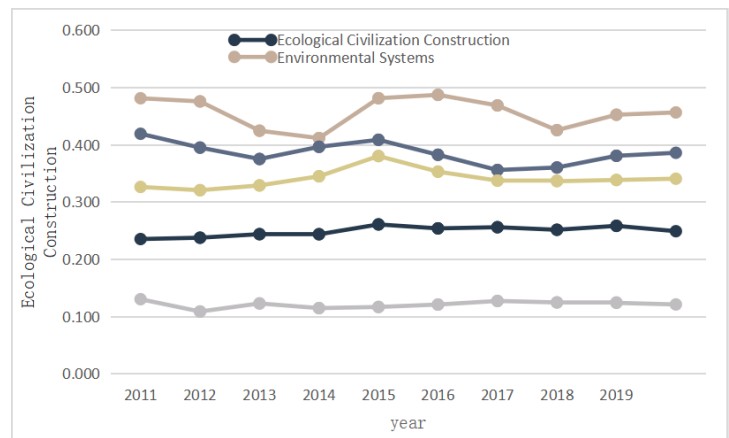

**Figure 4.** Temporal evolution pattern of ECC.

4.1.2. Analysis of the Spatial Evolution Pattern

Figure 5 shows the spatial framework of UGD in the YRD in 2010, 2015, and 2019. The cities with a higher green development level in 2010 were Nanjing, Suzhou, Hangzhou, and Shanghai, among which Shanghai has the highest level of 0.63. The western inland region of the YRD is where most of the cities with a lower level of green development are located. Among these, Lu'an has the lowest level of 0.18, indicating that there was a severe bifurcation of the UGD level among the cities in the YRD. In 2019, the UGD level in the high development zone comprising Nanjing, Wuxi, Shanghai, and neighboring cities, is still higher than that in other regions of the YRD. Shanghai scored the highest, at 0.63, followed by Wuxi and Nanjing, both of which scored 0.48. Huainan City, which had the lowest score of 0.19, is in the low UGD zone, which is primarily in the northwest of the YRD. This has shown that over the last ten years, the central–eastern cities in the central–eastern region

have shown more rapid green development. Overall, from 2010–2019, the level of UGD in the YRD was on an upward trend, but there were pronounced regional differences. ① With a high degree of UGD and a potent siphon effect, the eastern area, concentrated in Shanghai, Wuxi, and Nanjing, attracts top-notch technical personnel and market resources from the YRD. ② Most of the central region's cities, with Hefei, Wuhu, and Hangzhou serving as the region's hubs, are at a middle to high degree of green development. Their growth is aided by the "edge effect" of the metropolis. The center cities' ability to cluster green development has improved and their growth potential has increased more rapidly with integrated development of the YRD. ③ The northwestern part of the YRD is further away from the high growth urban agglomeration, and, therefore, the UGD was relatively slow.

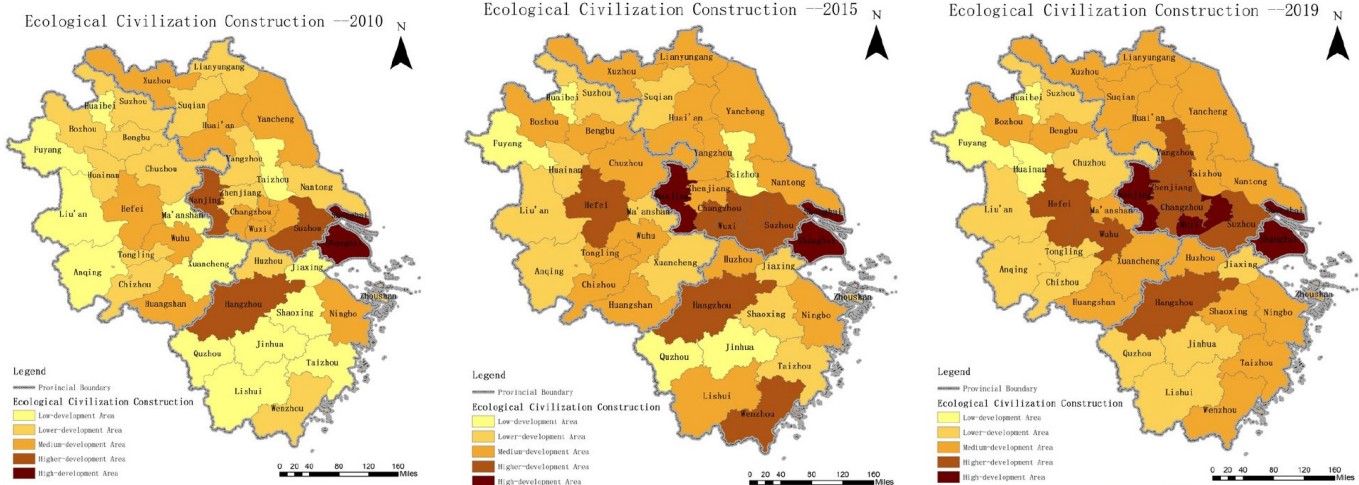

**Figure 5.** Spatial evolution pattern of UGD.

Figure 6 shows the spatial pattern of the level of ecological civilization in the YRD in 2010, 2015, and 2019. From 2010–2019, the overall ECC level is in a transitional pattern, with Nanjing, Shanghai, and Hangzhou forming three high level zones. This has then driven the development of ECC in surrounding cities to transition outward and drive the development of ECC in the YRD. From the perspective of provincial administrative regions, the level of ECC in the YRD is Shanghai > Zhejiang Province > Jiangsu Province > Anhui Province. In the three time cross-sections, in 2010, there are 23 cities in the lower level and low-level zone, accounting for 56.1%, which were predominantly concentrated in the northern part of the YRD. There were 13 cities in the middle level zone, accounting for 31.7%, which were mainly concentrated in the southeastern part of the YRD. There were five cities in the higher level and high-level zone, accounting for 12.2%, primarily in the region's central–eastern portion. In 2019, the number of lower level and low-level cities gradually decreased to 17, accounting for 41.5%, which were mainly concentrated in the northwest inland region. The number of medium level cities decreased to eight, accounting for 19.5%. Higher level and high-level cities accounted for 36.6%, an increase of 24.4% from 2010, which were mainly concentrated in the southeast coastal region. Figure 7 shows that the level of ecological civilization has had a more pronounced increase in the cities around Shanghai, Hangzhou, and Nanjing. Meanwhile, the cities further away from them have improved more slowly. The UGD and ECC level was positively correlated and the higher the level of UGD, the higher the level of ECC in cities. In terms of spatial distribution, the UGD and ECC of cities in the southeast coastal territory was more elevated than that of the entire northwest inland territory.

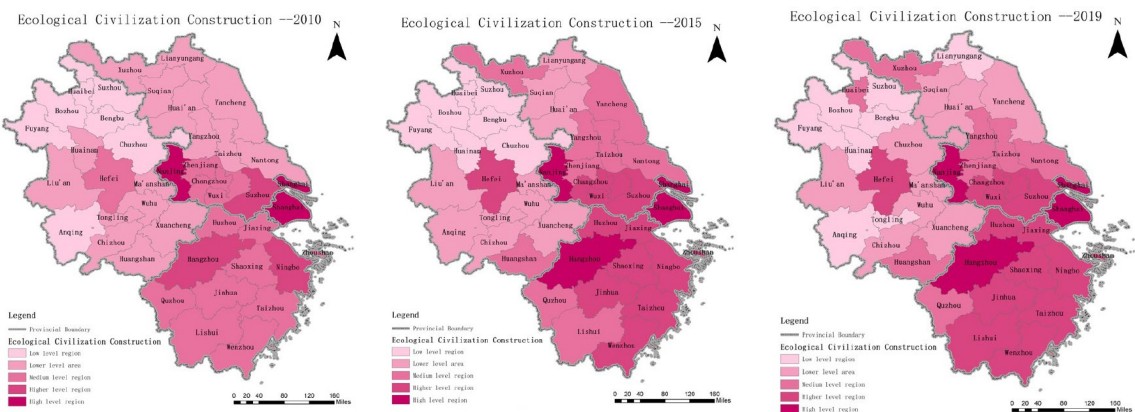

**Figure 6.** Spatial evolution pattern of ECC.

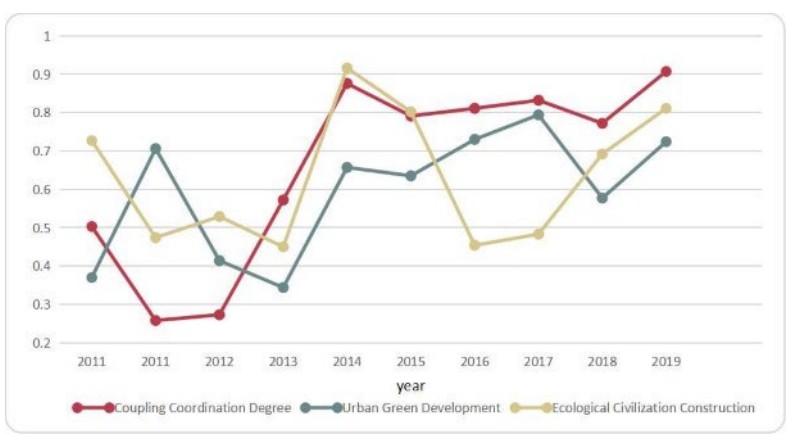

**Figure 7.** Temporal trend of coupling coordination.

### 4.2. Coupling and Coordination Analysis of UGD and ECC

4.2.1. Temporal Trends of Coupling Coordination

The degree of coordination of UGD and ECC in the YRD from 2010 to 2019 was measured using a coupled coordination degree model, and its development type was analyzed. Figure 7 shows that the degree of UGD in the YRD increased from 0.37 in 2010 to 0.724 in 2019. The degree of ECC increased from 0.727 in 2010 to 0.811 in 2019, and the degree of coupling coordination continued to increase from 0.503 in 2010 to 0.907 in 2019. All three of these indicators have been rising and fluctuating over the last ten years. This has shown that the level of collaboration between green development and ECC in cities in the YRD is still in the exploratory stage and is likely to improve in the future. A further indication that national policies have had a substantial impact on the coordinated advancement of green cities and ecological civilization is the timing of the change in coordination type. This is compatible with the point in time of China's 13th Five-Year Plan. The early stage of UGD (2010–2013) was in the exploration stage, when the UGD trend was more volatile and the impact on the ECC was at a lower level of coordination with the ECC. During this, many researchers compared the UGD situation in China and internationally and developed an UGD path applicable to China's national conditions [58–60]; the middle and late stage of UGD (2013–2019) belongs to the stage of fluctuation and rise, proposing "fostering green sustainable growth and constructing a beautiful China", when the road to UGD has become mature, and the influence and coordination of ECC have been greatly enhanced. However, the level of ECC dropped substantially because of strengthening of environmental protection. This has led to the increase in enterprise investment in pollution control, including measures such as shutting down production and rectification. Although the UGD and ECC coordination degree spans four stages from 2010 to 2019, the degree

of coupling coordination between the two systems still has not reached a high level of equilibrium. This is because there is still considerable space for improvement in the degree of coupling coordination between the two.

### 4.2.2. Coupling and Coordinating Spatial Patterns

The spatial distribution of UGD and ECC coordination in the YRD in 2010, 2015, and 2019 is shown in Figure 8. The YRD's general degree of cooperation has increased from 2010 to 2019. In 2010, 70.73% of the cities in the YRD were in severe dissonance and on the verge of dissonance. As of 2019, 60.97% of the cities in the YRD were in primary coordination and quality coordination. There are still substantial differences among the cities in the YRD in terms of coupling coordination. Generally, the southern YRD has a higher level of cooperation than the northern YRD, and the eastern coastline zone has a higher level of coordination than the western interior region. Analyzed from the perspective of provincial administrative regions, the order of UGD and ECC coordination is Zhejiang Province > Jiangsu Province > Anhui Province. The UGD and ECC high-level city cluster consists of Shanghai, Nanjing, and Hangzhou, and has played a key role in advancing ecological civilization and green urban development as the YRD's core region.

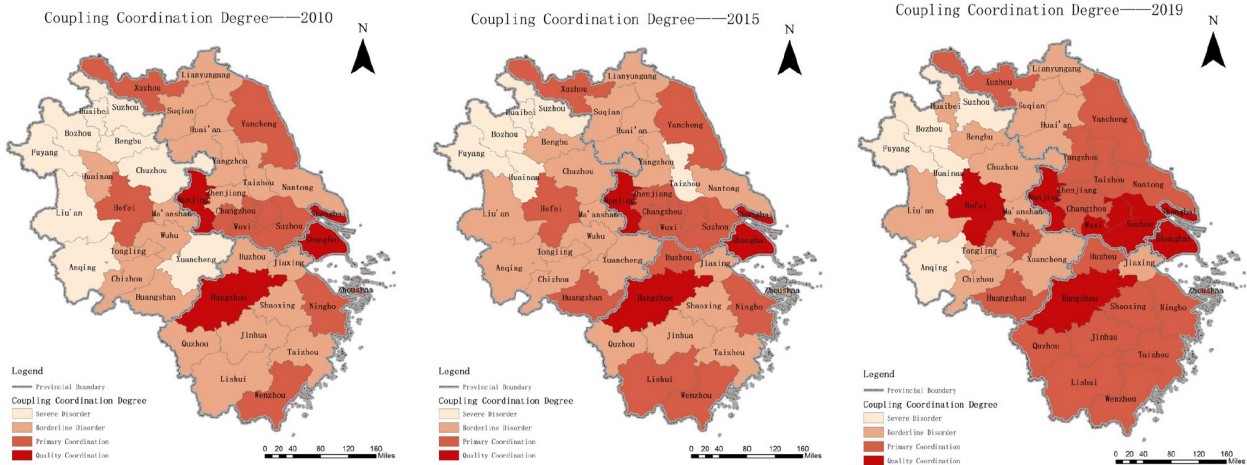

**Figure 8.** Spatial pattern of coupling coordination.

### 4.3. Analysis of Spatial and Temporal Differences between UGD and ECC

This study has examined the effects of UGD and its equipment on ECC and the effects of ECC and its equipment on UGD, from 2010 to 2019, to clarify the mechanism behind the connection between UGD and ECC. Table 2 presents the specific modeling variables and findings. All the models have a high level of explanation and can explain the link between UGD and ECC.

**Table 2.** GTWR model calculation results and parameters.

| Dependent Variable | R2 | Bandwidth | Sigma | AICc | Spatio-Temporal Distance Ratio | Residual Squares |
|---|---|---|---|---|---|---|
| UGD | 0.933704 | 0.114996 | 0.0191119 | −1810.77 | 0.373068 | 0.149758 |
| ECC | 0.970262 | 0.114996 | 0.023251 | −1724.87 | 0.268765 | 0.221651 |

### 4.3.1. Time Variance Analysis

The effects of UGD and its equipment on the ECC at the timeframe are shown in Figure 9a−d. UGD gradually tends to have a positive impact on the ECC, and the intensity of that impact has gradually increased. Social life, industrial production, and economic growth have all had an impact on UGD levels, and when those levels have been reached, the ECC will improve.

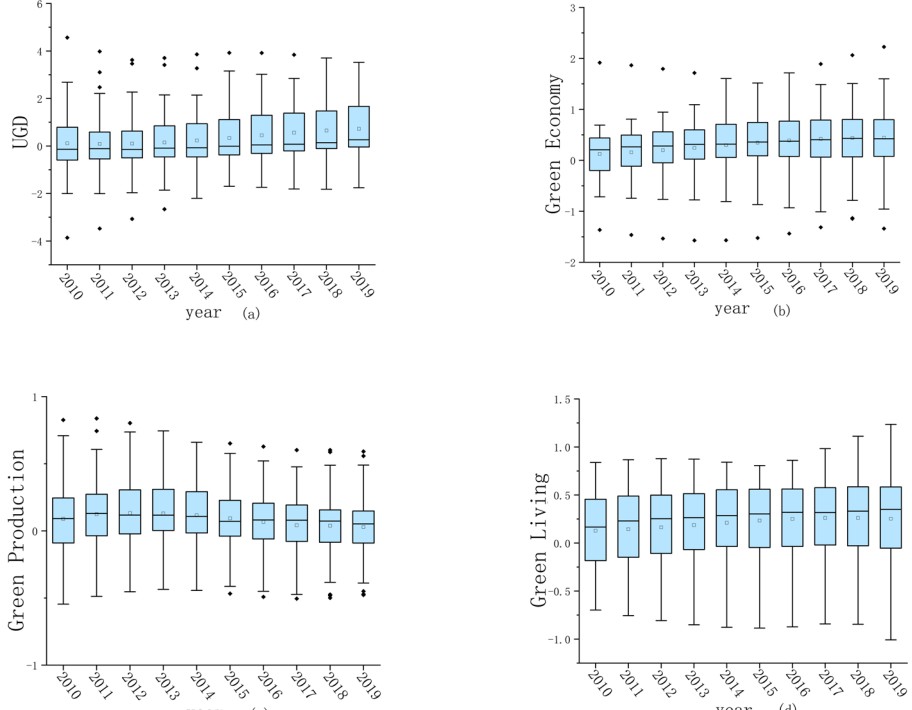

**Figure 9.** Analysis of the temporal variability of UGD. Where (**a**) indicates the level of influence of UGD on the ECC; (**b**) indicates the level of influence of green economy on the ECC; (**c**) indicates the level of influence of green production on the ECC; (**d**) indicates the level of influence of green living on the ECC.

① As can be seen in Figure 9b, the positive effect of the green economy on the level of ECC has shown a steady upward trend. This is primarily closely connected to government support and changes in the market environment, developing low-carbon and environmentally friendly development, promoting sustainable resource use, putting energy conservation and emission reduction efforts into practice, actively developing the circular economy, and working to create a green life production model.

② From Figure 9c it can be seen that green output has a first-rising positive impact on the level of ECC. It then falls and finally tends to stabilize the trend, predominantly because the continuous development of industrial production has driven the construction of the urban economy and urbanization level. While indirectly supporting the development of urban ecological civilization, this also serves to bolster green production and realize a society where production and ecology coexist peacefully. However, as production levels have risen steadily, they have also caused severe environmental damage and put undue strain on natural resources, lowering the level of ECC.

③ Figure 9d show that people's adoption of a green lifestyle has had a favorable impact on the level of ECC that is more consistent and on the rise. At a meeting of the Political Bureau of the 18th Central Committee of the CPC Central Committee Standing Committee in 2013, Chinese Leader Xi Jinping made an important speech in which he "advocated green lifestyle," that is, a green and sustainable way of living for people, as a means for socialist nations to promote the ECC and to steadfastly carry out the requirements of the new development concept, which is more conducive to promoting the ECC way of life.

The effects of ECC and its equipment on UGD over time are shown in Figure 10a−e. Overall, the impact of ECC on UGD has changed from negative to positive, and the intensity of the impact has gradually increased.

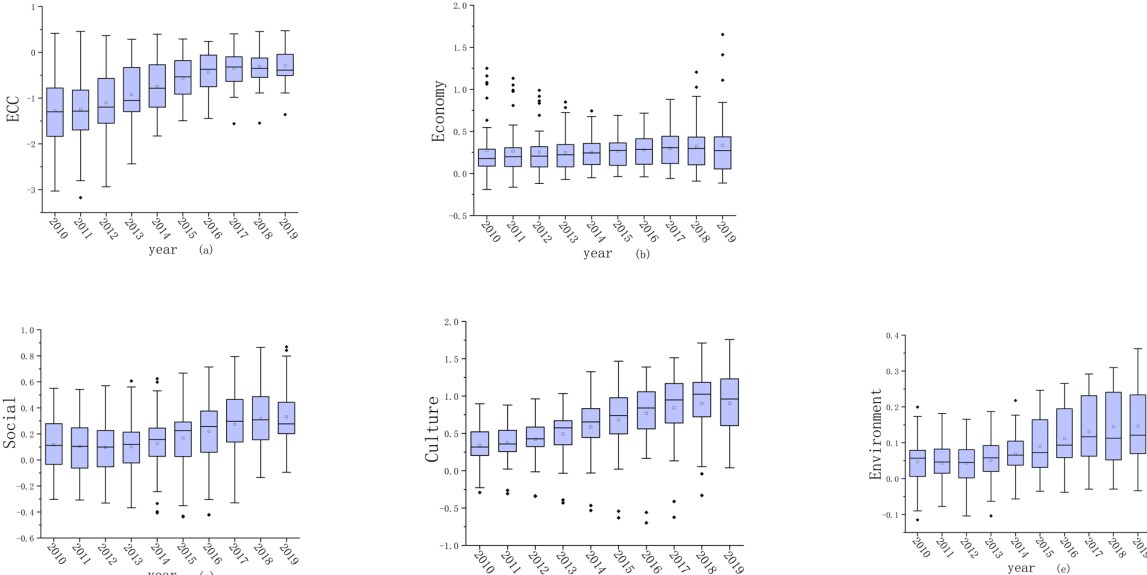

**Figure 10.** Analysis of the temporal variability of ECC. Where (**a**) indicates the level of influence of ECC on UGD; (**b**) indicates the level of influence of economic system on the level of UGD; (**c**) indicates the level of influence of social system on UGD; (**d**) indicates the level of influence of cultural system on UGD; (**e**) indicates the level of influence of environmental system on UGD.

① As can be seen in Figure 10b, the trend of the economic system's positive influence on UGD is upward, implying that as the economy's level of technological development rises, the strategy for developing the digital economy has been gradually optimized. As the strategic tasks for UGD have been developed and proposed, the rise in digital development of economy quickly becomes the primary influencing factor for advancing UGD. Zhu et al. (2022) noted that the digital economy has had a highly positive impact on the improvement in UGD level and has turned into an important driving force for urban green transformation, and the degree of influence is gradually increasing.

② From Figure 10c it can be seen that the social system's beneficial impact on UGD is increasing and the influence of social development on UGD declined slowly before 2013. This is because of development of urban productivity in the YRD before 2013, which caused more pressure on the environment and inhibited the speed of UGD. After 2013, China's urban economies have increasingly transitioned toward green and environmentally friendly growth. To advance the UGD process, it is vital to include the idea of ecological civilization into all facets of the urbanization development process and to follow a new low-carbon and environmentally friendly path for urban construction.

③ Figure 10d show that there has been a gradual upward trend in the positive influence of cultural systems on UGD. This is closely related to the strong government support for education schools and the development of emerging technologies. Green sustainable development of cities can be realized with the expansion and support of green culture, which generally has a positive guiding influence on this process. Without intensive cultivation of people's green cultural consciousness, sustainable urban development cannot be accomplished. The sustainable development of cities cannot be achieved without the extensive cultivation of people's green cultural awareness. To strengthen the excavation and inheritance of cultural resources, actively cultivate a green culture and implant the cultural gene of green development, so that the concept of green water and green mountains as the silver mountain of gold becomes a universal concept in today's society, further promoting the intrinsic power of green development in the city.

④ As can be seen in Figure 10e, the influence of environmental systems on UGD is similar to that of culture, both of which have an annual rising trend, was mainly related to

government policies and industrial structure. Eco-environmental improvement is the fundamental driving force and necessary measure for UGD. Hu [61] has highlighted that the imbalance in the ecological environment is the fundamental cause of inhibiting UGD and restricting urban socio-economic development. As a result, in the cause of advancing the UGD process, the YRD has worked hard to enhance the ecological environment's quality by expanding investments into high-tech and green sectors.

### 4.3.2. Spatial Variation Analysis

The average of the yearly impact coefficients of the interactions between the UGD and ECC in the YRD from 2010 to 2019, as shown in Figures 11 and 12. The parametric results are based on the GTWR model. As shown in Figure 11, UGD has improved the level of ECC in the southeastern local-level cities in the YRD but inhibited the level of ECC in the northwestern prefecture-level cities. This has demonstrated that the level of green development varies substantially amongst some of the cities in the YRD, with Shanghai, Nanjing, and Hangzhou acting as the region's hub and outward extensions. Given the booming seaside economy, rising living standards, and the green industry's dominance of the industrial structure, the southeastern cities are growing quickly. The cities in the northwest are expanding slowly because of the slow growth of the inland economy, with heavy industry making up the bulk of the main industrial structure. This has resulted in the slow growth of the local ecological civilization. To encourage coordinated growth among UGD and ECC, it is therefore required to design various strategies in accordance with the differences in geographic location, economy, industrial composition, and urban infrastructure.

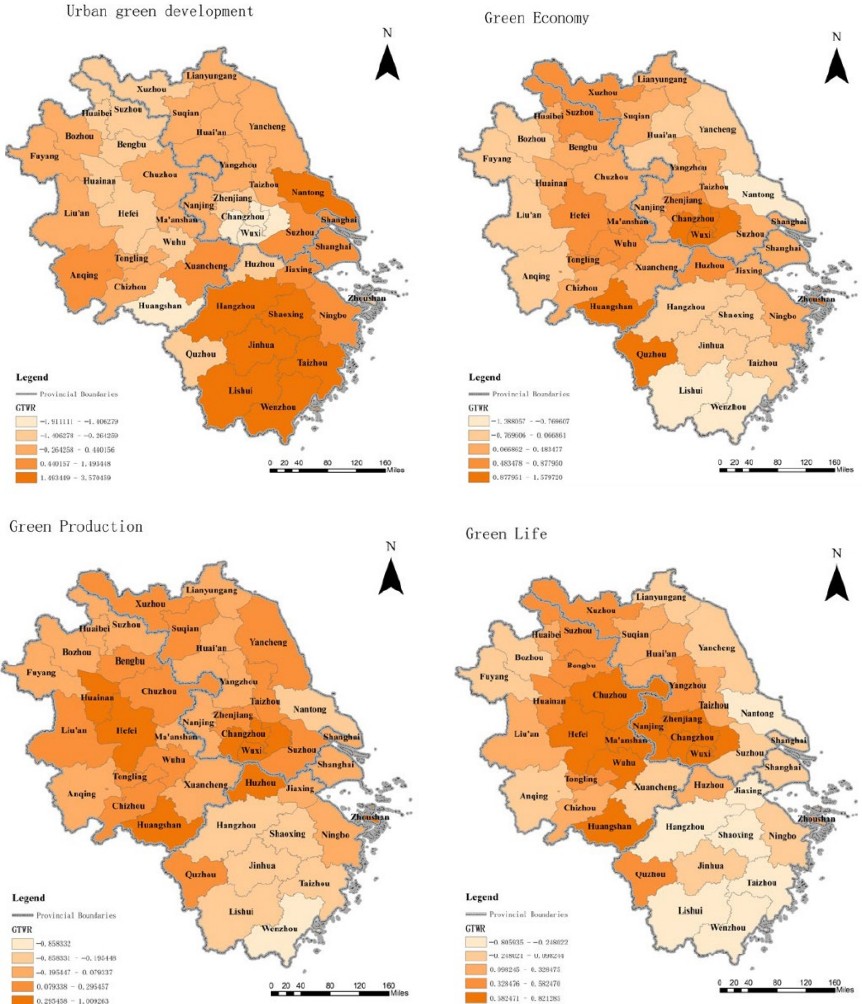

**Figure 11.** Spatial variation analysis of UGD.

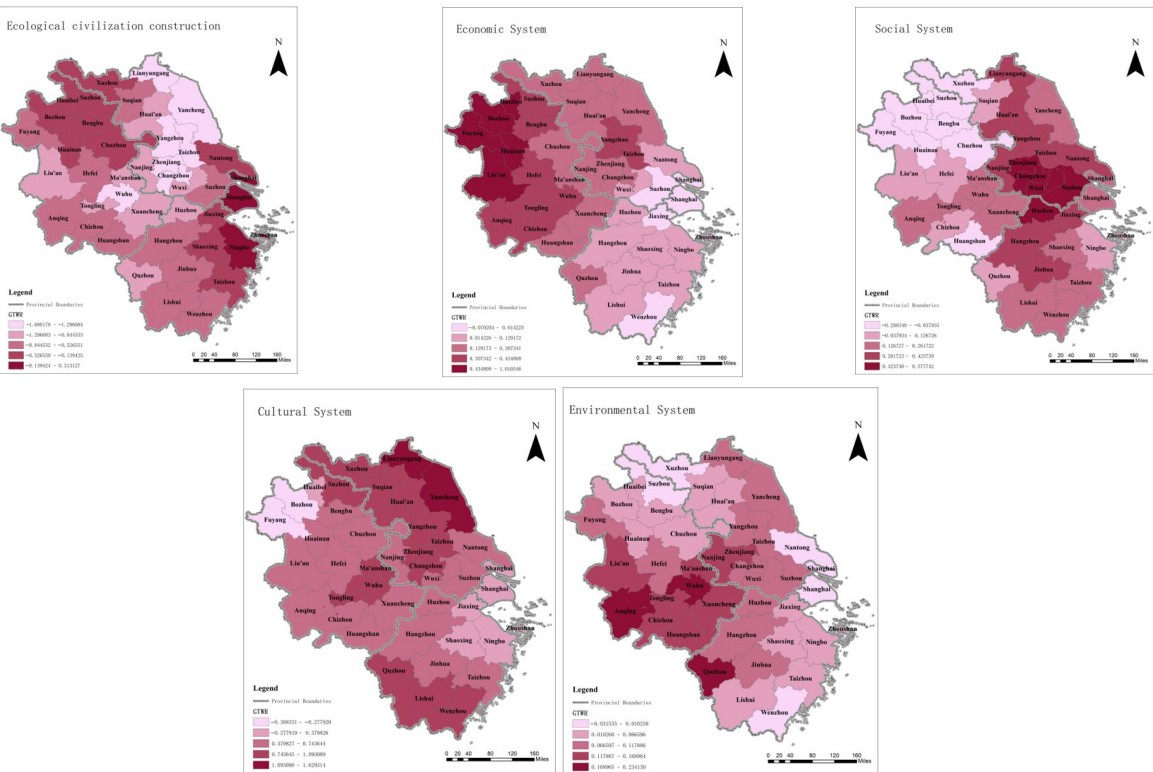

**Figure 12.** Spatial variation analysis of ECC.

① The level of green economic development has brought positive effects for most cities within the YRD, and only a few cities have negative regression coefficients. Huangshan City has the highest absolute value of coefficients of the level of green economic development, followed by Wuxi City, while Hangzhou City has the lowest absolute value. There are two scenarios when there is a correlation between the rate of green economic growth and the amount of ECC. In the central cities of the YRD, such as Huangshan City, Wuxi City, Changzhou City, and Quzhou City, the degree of influence of the extent of green economic progress on the extent of ecological and environmental protection is high. However, the ecological and environmental protection standards of these cities are at a low–medium level. To improve the ecological and environmental protection levels of these cities, the focus should be on improving the local level of green economic development. The extent of green economic progress, such as that of Hangzhou, which is a high level ECC area, has a low impact on the level of ECC but has a high influence on the nearby cities' levels of green economic development. This suggests that the city should actively promote the extent of green economic development in the surrounding cities to substantially increase the level of ecological civilization in the area.

② Green production has brought a greater positive impact to the northern cities of the YRD, while it has a smaller impact on the southern cities. The city with a positive coefficient and the largest absolute value of green production level was Wuxi, followed by Changzhou. Meanwhile, the city with a negative coefficient and the largest absolute value was Wenzhou. Green production is closely tied to the impact of the green economy on the degree of ECC. The progress of green production is the foundation for encouraging economic growth and a crucial way of improving the ECC.

③ Green living has a greater positive impact on the central YRD. Meanwhile, it has a smaller impact on the YRD's southern cities. Here, the coefficient is effective and the highest in absolute amount is Chuzhou City, followed by Maanshan City, and where the coefficient is negative and the highest in absolute amount is Taizhou City.

Figure 12 shows that the ECC has improved green progress of eastern coastal cities while restricting the green development of western interior cities in the YRD. Environmental restrictions and urban economic development are the main causes of this divergence [62]. Heavy industry is the primary driver of urban economic building in the western cities of the YRD. This is the main cause of aggravation of the eco-environment and the limitation of UGD. In contrast, the eastern cities have more light industries than heavy industries. This means that there is less environmental interference, ensuring the sustainable green progress of the cities.

① The progress of the economic system has brought positive effects to most cities in the YRD, and only a few cities have negative regression coefficients. Fuyang City has a positive regression coefficient and the highest absolute value, followed by Bozhou City. Meanwhile, Shanghai City has a negative regression coefficient and the highest absolute value. Based on this analysis, economic growth is the primary driver of the green progress of cities like Fuyang and Bozhou and has a stronger impact on it. Economic progress is an important driving force of UGD [31]. Therefore, focusing on improving the local economic construction is a necessary means to enhance UGD. Meanwhile, in cities such as Shanghai, which is different, economic development has a smaller influence on UGD. These cities have a higher level of green development, which has a smaller influence on the nearby cities. Given that the green development level of such cities is greater and influences the other cities, it is advisable for Shanghai or other high-development cities to serve as the region's economic hub to drive the green development level of all the nearby cities.

② While restricting the green progress of the cities in the northwestern part of the YRD, the social system development has increased the level of UGD in the central–eastern part of the YRD. Here, the regression coefficient is positive and has the largest absolute value in Huzhou City, followed by Wuxi City, and is negative and has the largest absolute value in Fuyang City. The primary factor is that urban growth is more advanced and social system progress is at a relatively high level in the central–eastern region of the YRD. This has formed a high-level agglomeration of social system development, which has a higher impact on the UGD of the region. Meanwhile, in the northwest, there has been less urban construction and the level of social development is relatively low. This has formed a low-level agglomeration of urban clusters, which has a smaller impact on the UGD of the region. Therefore, the high-level area should drive the low-level area to develop gradually to the northwest, forming a high level of regional development, to raise the YRD's overall UGD standard.

③ Most cities in the YRD are positively influenced by the development of cultural systems. Only a few cities at the edges have been negatively influenced, among which the largest positive influence coefficient is Yancheng City, followed by Lianyungang City. Meanwhile, the largest negative influence coefficient in absolute value is Fuyang City. Most of the cities with negative influence coefficients are located at the edges of the YRD. One is in the southeast coastal area, and the other is in the northwest inland area. There are differences in geographic and economic factors in the development of these two regions. The economic advance of the southeast coastland has been more rapid and the level of UGD is higher. Meanwhile, the cultural construction has had less influence on the UGD of the region. The economic advance of the northwest inland region is slower, and the extent of social development is lower, leading to the cultural construction of the region being less strong with less influence on the UGD. Therefore, while strongly developing the economic level of the region, the social and cultural construction is also essential, and these have an essential promotional role the UGD of the region.

④ The number of positive impacts from environmental systems on cities in the YRD is 90.24%. Only 9.76% of cities were negative, among which Anqing, Wuhu, and Quzhou had the largest positive impact coefficients. Meanwhile, Wenzhou, Shanghai, and Nantong had negative impacts and the largest absolute values. The influence of the

environment on the level of UGD in the YRD is divided into two situations. The first is that cities in the western part of the YRD, such as Anqing City and Wuhu City, have a high level of effect of the environment on the degree of UGD. However, their level of green development is low, indicating that to improve their level of green development, the level of local ecological environment construction should be improved. Cities including Wenzhou City and Shanghai City have a high degree of influence from environmental development on the influence of UGD level, which is relatively low. However, the level of green development of these cities is relatively high. This indicates that the environmental construction of the city is more outstanding and the green industry accounts for a larger proportion of the industrialization. These cities should actively drive the environmental construction of the neighboring cities.

## 5. Discussion

This paper takes the level of UGD and ECC in the YRD region as the research object, and theoretically analyzes the coupling relationship and driving mechanism between the two. UGD is evaluated in three dimensions: green economy, green production, and green life, while the construction of ecological civilization is evaluated in four dimensions: economic–social–cultural–environment, and a comprehensive evaluation system of UGD and ECC indicators is constructed. On this basis, the spatial and temporal evolution rules of the coupling and coordination of UGD and ECC, as well as the spatial and temporal differential evolution characteristics, are studied. Under the general direction of building a community of human destiny and realizing global green development, the research in this paper is crucial to exploring the path of sustainable development.

First of all, there are few studies on the coupling relationship between UGD and ECC, and studies focusing on the spatial and temporal characteristics and interactive coupling theory are scarce. In view of this, this paper analyzes the influencing factors between UGD and ECC from the connotation concept of the two and studies the interactive relationship between them. Through the study, we find that the existing literature mostly adopts a single dimension or a single indicator to measure UGD and ECC, such as green development efficiency [63] and eco-efficiency [64], but green development efficiency and eco-efficiency are only analyzed from the input–output perspective of production, ignoring the development momentum and urban structure. Similarly, there are studies that construct indicator systems to comprehensively evaluate UGD and ecological civilization construction, such as the PSR model [65] and the DPSIR model [66], which are more comprehensive and systematic in evaluating research objects through indicator systems. It is noteworthy that the existing studies are mainly based on the definitions of green development and ecological civilization in national policy documents and evaluation standards, i.e., "resource use—environmental management—environmental quality—ecological protection -Growth quality", "resource use—environmental protection—annual evaluation results—public satisfaction—ecological and environmental events" [67,68]. In this paper, the evaluation system of urban development level is based on the "green model", which selects indicators from economic–production–living dimensions, and for the construction of ecological civilization, this paper selects indicators based on "economic-social-cultural-environmental" multi-dimensions. This paper enriches and improves the research on the evaluation of green development level and ECC in cities.

Second, most previous studies have used green development efficiency and eco-efficiency to study the level of UGD and the level of ECC. These approaches treat UGD and ECC as two independent variables, ignoring the interaction mechanism between UGD and ECC [69]. This study uses a coupled coordination model to further investigate the interaction between the two, and from the analysis of the results, many sustainable development efforts have been conducted in recent years in Shanghai, Nanjing, and other cities to raise the level of green development and ecological civilization in cities. Wei [36] has highlighted that to foster sustainable urban development, we need to constantly adhere to the fundamental principle of peaceful coexistence between humans and the natural world.

This is because sustainable urban development is a necessary and intrinsic component of the ECC. With reference to the combination of the 2019 *Zhejiang Provincial Statistical Yearbook* and *Anhui Provincial Statistical Yearbook*, the "usage of energy per 10,000 yuan of GDP" in Zhejiang Province is 0.26 tons of standard coal, which is 38.68% less than that of Anhui Province. The green growth of Zhejiang cities is less reliant on resource use and places less strain on the environment. This has resulted in a higher level of coordination amongst UGD and ECC. To improve environmental quality, the governments of Zhejiang Province and southern Jiangsu Province have made substantial investments in the development of resource-efficient and environmentally friendly companies [54]. However, the economic growth of Anhui Province and the northern region of Jiangsu Province is primarily dependent on the export of resources and the building of infrastructure, which to some extent compromises ecosystem integrity. The transfer of high-contamination and high-discharge industries to the region has exacerbated the environmental contamination in these areas and places stress on the ecological environment. Several initiatives have been developed by the local authorities to address this issue. Anhui Province's "12th Five-Year Plan" proposed the "Notice of the Anhui Provincial People's Government on the Issuance of the "13th Five-Year Plan" for the implementation of energy conservation and emission reduction", the "13th Five-Year Plan" proposed the "Anhui Province The 13th Five-Year Plan for Ecological Protection and Construction" and the "Jiangsu Province Ecological Protection and Construction Plan (2014–2020)" published by Jiangsu Province in 2015. The promotion of effective coordination between UGD and ECC has been strongly supported by each of these strategies.

Third, this paper provides a new perspective for the spatio-temporal analysis of UGD and ECC. Existing studies mainly use ESDA [53], SDA [70], and spatial autocorrelation models [71], ignoring the spatio-temporal heterogeneity of the degree of influence of different factors. This study examined the spatio-temporal non-stationarity of the influencing factors through GTWR. In order to conduct a more in-depth study, this paper considers regional differences in the level of UGD and ecological civilization construction. This paper considers the regional differences in the level of UGD and ECC. Analyzed from the time dimension, the impact of green economy and green life on the construction of ecological civilization in cities in the YRD region during 2010–2019 is positive and has an increasing trend, indicating that the growth of the green economy is essential for advancing the ECC and is a key factor in raising the degree of ecological civilization [72]. According to the 14th Five-Year Plan, to create a society that is prosperous in all respects, we need to create a green economy, encourage the advancement of science and technology, and continue the transformation and modernization of important sectors of the economy. Key links and key industries, all-round adjustment and optimization of transformation, resolutely fight the battle of pollution prevention and control and construction of the battle. The impact of green production on the construction of ecological civilization is in an unstable trend, the reason is that production is a double-edged sword; vigorous development of production will promote the economic development of the city, but there is the possibility of causing excessive pressure on resources and inhibiting the construction of ecological civilization. Therefore, exploring green development is the way to achieve the building of a community of human destiny. The impact of ecological civilization on the green development of cities is also tending to increase positively, illustrating that the 2019 Master Plan of the YRD Eco-Green Integrated Exploitation Demo Area has accelerated UGD's advancement in the YRD and, to a certain extent, the creative and coordinated growth among UGD and ECC. All four subsystems—economic, social, cultural and environmental—have a positive influence on UGD, which is in line with the findings of Jiexi Zhu [46] and Hu [61] and others. From the spatial dimension, the level of green development and the construction of ecological civilization in cities is centered on high-level urban agglomerations, such as Shanghai and Nanjing–Hangzhou, with the closer the city to the high-level urban agglomerations, the higher the level of development, and the further the distance, the slower the level of development. Based on this, regional governments have introduced corresponding

measures to promote the level of green development and ecological civilization in cities, such as the "Strengthen the protection of ecological environment in an all-round way and resolutely fighting the tough battle of pollution prevention." published by Anhui Province in 2018. This was undertaken with the intention of reducing environmental pollution from secondary industries that produce high emissions and pollution levels. They also included a proposal to actively promote the growth of green industries, and protection of the environment in cities is rising along with people's economic levels and the desire for high-quality environments. This is advantageous for the enhancement of ecological and environmental quality. Moreover, the use of big data networks has decreased paper use and enhanced environmental quality.

Fourthly, and finally, the outlook and shortcomings of this paper. In the context of global common construction of a human destiny community, UGD and ECC become important driving forces to promote the construction of high-quality cities, study the coupling coordination and spatial difference of UGD and ECC, clarify the direction of urban development, drive the development of neighboring cities by high level cities, and eventually drive the strategic goal of high quality development of national cities by urban clusters. Due to the limitation of data availability, this paper studied the coupled and coordinated relationship and differences between UGD and ECC during the 10 years from 2010–2019, and seldom considered some micro-level factors when constructing the comprehensive evaluation system of ECC and UGD, and how to accurately measure ECC and UGD remains the main research direction in the future.

## 6. Conclusions and Recommendations

### 6.1. Conclusions

Promoting green and sustainable development of the city is an important element to resolutely fight the battle of pollution prevention and control, crack urban diseases, improve the city's taste and build a green city, and achieve the scientific development of information technology in the city at a higher level. Existing studies have focused primarily on UGD and ECC's qualitative relationship with the coordination relationship inside a single system, while disregarding UGD and ECC's coordination relationship and inner mechanism. Therefore, the CCDM and GTWR models were combined to examine the coupling and coordination effects and spatio-temporal heterogeneity among UGD and ECC of prefecture-level cities in the YRD from 2010 to 2019.

The results of the study have shown that: ① From 2010 to 2019, the level of UGD gradually increased, and the priority of urban development shifted from actively growing industrial level and driving economic building to insisting on ecological conservation and driving green development. Eastern coastal cities of the YRD have more green development than those in the western interior. The rate of ECC has indicated a slow but steady increase from 2010 to 2019. Meanwhile, the importance of the influence of economic system and cultural system on the ECC is higher among all evaluation indicators. The ECC in coastal cities in the southeastern YRD is better than other cities. ② The level of coordination among UGD and ECC inside the YRD fluctuated upward from 2010 to 2019. It underwent the change from lagging UGD to lagging ECC, evolved from a phase with no correlation to a primary coordinated phase. Meanwhile, the degree of coordination in the eastern coastal cities in the YRD has become higher than that in the western inland cities. ③ The UGD and ECC interaction tends to be significant, according to the GTWR model. The UGD and ECC's adverse effect between YRD is concentrated in the northwestern inland cities, while the positive effect of UGD and ECC is concentrated in the southeastern coastal cities. This has indicated that UGD and ECC complement each other, and UGD is the fundamental driving force of ECC, while ECC is the goal and orientation of UGD.

*6.2. Policy Recommendations*

To improve the degree of coordination among UGD and ECC and considering the differences in geographic location, resource endowment and urban development patterns, this study has proposed the following recommendations:

The overall level of green development in the country should continue to be improved, and "development" and "green" should be coordinated and advanced. Effective handling of the harmony between humans and nature should always be prioritized in the specific urbanization process economic development. Chinese President Xi Jinping has said everyone should abide by the green development tenets that "green water and green mountains are the silver mountain of gold", and "harmony between humans and nature" is central to it. He said will fully implement the spirit of the 20th Party Congress in urban production, people's lives, and ecological protection. We need to pay full attention to the unbalanced spatial characteristics of green development, and governments at all levels should actively take measures to effectively narrow the gap between the economic development level and industrial structure of each region.

The focus should be on improving the level of green production in the city to achieve the purpose of substantially improving the level of green development. In the Opinions on Promoting Green Development in Urban and Rural Construction issued by the State Council, it is stated that urban planning and design should adhere to resource recycling, green sustainable development, regeneration, and continuous recycling, implement green transformation of traditional industries, develop green industries, and build a sound green industrial system. In its socio-economic development, China insists on implementing the new basic principles of circular economy, i.e., reduction, resourceization, resource repositioning, resource replacement and harmless storage, realizing intensive recycling of resources, reducing the solid waste that may be generated, promoting the coordinated development of green regions, and developing new green industries and models to contribute significantly to the realization of building a community of human destiny.

The areas with higher levels of green development should be used as the center to radiate the surrounding areas and drive the development of the surrounding areas. Regions with lower levels of green development can then learn from the development model of regions with higher levels. Meanwhile, inter-regional cooperation should be guided to benefit from complementary advantages and narrow the differences between regions. Depending on the environment and resource bearing capacity, the density of current development, and the potential for future development, each region should gradually form the main function positioning division of labor to form a benign interaction between cities and coordinate development.

**Author Contributions:** Conceptualization, C.D.; methodology, X.H.; validation, formal analysis, X.H.; data collation, Y.W.; writing—original manuscript preparation, X.H. All authors have read and agreed to the published version of the manuscript.

**Funding:** This paper was supported by the National Natural Science Foundation of China (No. 71773117, 71903183); Major Projects of the National Social Science Foundation of China (No. 18ZDA066); Fees for basic scientific research of Chinese Academy of Surveying and Mapping (No. AR2207).

**Data Availability Statement:** The authors do not have permission to share data.

**Conflicts of Interest:** The authors declare that they have no known competing financial interest or personal relationships that could have appeared to influence the work reported in this paper.

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
