# Peer review of "Coupled and Coordinated Analysis of Urban Green Development and Ecological Civilization Construction in the Yangtze River Delta Region"

_sustainability, doi:10.3390/su15075955_

Round 1

Reviewer 1 Report

Dear authors,

I have been pleased to have an opportunity to read your paper. I think that you have a very interesting theme but your paper needs to be improved:

1. You have to include investigations made in other parts of the world because ecology does not have boundaries and can not be viewed only throughout exploration of Chinese experiences. For ecological explorations we are all in the same planet and therefore we have to explore what scientists from other countries and continens are saying about urban green development. So please, refer to the worldwide experiences and references.

2. Discussion as a chapter of your paper needs to be implemented.  It will help in a clarity of your structure and clearness of the paper.

3. You have technical problems within the paper in references, please check them all. For example, this is one of mistakes - lines 123-124 which are citing Huang et al. is not 30th reference but 29th and there are more of these mistakes.

Relying on the GWR model, Huang et al. [30] examined the spatial relationship and regional variation between the Yangtze River Economic Zone economic, environmental, and resource systems and levels of ecological civilization. (moreover this sentence is a bit strange, please reconsider it)

29. Huang, Z.; Wang F.; Cao W. Exploration and prediction of factors influencing the level of ecological civilization in Yangtze 861 River Economic Belt--based on combined VAR and GWR-BP neural network model. Econ Geogr 2020,40(03):196-206.

 4. Can you please explain this (lines 581-584)? 

It is important to encourage the active cultivation of green culture, the implantation of the cultural gene for green development, the idea that green water and green mountains are the silver and gold of today's society, and the further promotion of the inner force of green development of the city.

5. Chapter Policy Recommendations is full of phrases WE - who is that? Chinese government, Chinese people or maybe not only Chinese government and population?

If that is a positive answer and would recommend you once more to look out of the box and try to make a broader picture when talking about ecology.

Reviewer 2 Report

I think this is a good job. The topic and research work are very interesting. I just have some minor comments. On the section of introduction, the academic contribution of this paper to the research topic should be elaborated more clearly. I suggest the authors to add more texts to showcase the academic contribution of this paper.

Reviewer 3 Report

Thanks a lot for the opportunity to review the manuscript entitled "Coupled and Coordinated Analysis of Urban Green Development and Ecological Civilization Construction in the Yangtze River Delta Region" (sustainability-2295030). This paper applied the coupled coordination degree model (CCDM) and spatio-temporal weighted model (GTWR) to analyze the relationship and heterogeneity between ecological civilization construction and UGD and ECC in each city in the Yangtze River Delta region. The paper could make potential contributions both from methodological and perspective level, there are, however, a few questions to be clarified:

1.Although the authors summarized the related previous works, it is suggested that the author should categorize and review them according to certain logic,  rather than just list them directly

2.Three contributions have been highlighted on page 3, while innovation point 2 and 3 seem to be similar and can be integrated.

3. The authors addressed that a new GTWR approach has been developed (on page 3), but on page 9 the author also pointed out that the spatio-temporal GTWR proposed by Huang Bo et al. [34], how do you clarify your methodological contribution?

4. The discussion (e.g., theoretical implications) of the findings, and limitations and future research directions were not pointed out in the paper.

Reviewer 4 Report

This study used the coupled coordination degree model (CCDM) and spatio-temporal weighted model (GTWR) to analyze the relationship and heterogeneity between ecological civilization construction and UGD and ECC in each city in the Yangtze River Delta region from 2010 to 2019. In general, the idea is clear, the structure is reasonable, the article has certain practical significance. Some suggestions:

1.      2.1 Study area: This part of the content is too cumbersome, and there is a considerable amount of content that needs to be elaborated in the intdoduction section as a basis for selecting this study for this study.

2.      Can Figures 2 and Figure 3 be combined?

3.      A total of 20 secondary indicators such as the rate of forest cover  and urban population density are used to construct an index system for measuring the ECC level in the YRD. What is the basis for selecting these indicators? Is there a collinearity problem between different indicators? Is it possible to repeat?

4.      Figure 4. Green living shows a trend of overall decrease. What's the reason? Is it credible?

5.      There are significant formatting issues in the article, such as the way references are labeled, the format of formulas, etc. It is recommended that the author carefully read the full text and carefully revise it.

Round 2

Reviewer 1 Report

Dear authors,

Thank you for making all changes that I suggested.

Kind regards